# Prognostic Implications of the Novel Pulmonary Hypertension Definition in Patients with Aortic Stenosis after Transcatheter Valve Replacement

**DOI:** 10.3390/jcm11154279

**Published:** 2022-07-22

**Authors:** Dionysios Adamopoulos, Stamatia Pagoulatou, Georgios Rovas, Vasiliki Bikia, Hajo Müller, Georgios Giannakopoulos, Sarah Mauler-Wittwer, Marc-Joseph Licker, Nikolaos Stergiopulos, Frédéric Lador, Stéphane Noble

**Affiliations:** 1Faculty of Medicine, Department of Medicine, Geneva University, 1206 Geneva, Switzerland; marc-joseph.licker@hcuge.ch (M.-J.L.); frederic.lador@hcuge.ch (F.L.); stephane.noble@hcuge.ch (S.N.); 2Department of Internal Medicine, Division of Cardiology, Hôpitaux Universitaires de Genève (HUG), 1205 Geneva, Switzerland; hajo.muller@hcuge.ch (H.M.); georgios.giannakopoulos@hcuge.ch (G.G.); sarah.mauler-wittwer@hcuge.ch (S.M.-W.); 3Laboratory of Hemodynamics and Cardiovascular Technology, Ecole Polytechnique Fédérale de Lausanne (EPFL), 1015 Lausanne, Switzerland; m.pagoulatou@hotmail.com (S.P.); georgios.rovas@epfl.ch (G.R.); vasiliki.bikia@epfl.ch (V.B.); nikolaos.stergiopulos@epfl.ch (N.S.); 4Department of Acute Medicine, Division of Anaesthesiology, Hôpitaux Universitaires de Genève (HUG), 1205 Geneva, Switzerland; 5Department of Internal Medicine, Division of Pneumology, Hôpitaux Universitaires de Genève (HUG), 1205 Geneva, Switzerland; 6Pulmonary Hypertension Program, Hôpitaux Universitaires de Genève (HUG), 1205 Geneva, Switzerland

**Keywords:** pulmonary hypertension, aortic stenosis, transcatheter aortic valve replacement

## Abstract

**Introduction**: Pulmonary hypertension (PH), traditionally defined as a mean pulmonary artery pressure (PAP) ≥ 25 mmHg, is associated with poor outcomes in patients undergoing a transcatheter aortic valve replacement (TAVR) for severe aortic stenosis (AS). Recently, a novel definition for PH has been proposed, placing the cut-off value of mean PAP at 20 mmHg, and introducing pulmonary vascular resistance as an exclusive indicator for the pre-capillary involvement. In light of the novel criteria, whether PH still preserves its prognostic significance remains unknown. **Methods**: The study population consisted of 380 patients with AS, who underwent a right heart catheterization before TAVR. The cohort was divided according to the presence of PH (n = 174, 45.7%) or not. Patients with PH were further divided into the following groups: (1) Pre-capillary PH ((Pre-capPH), n = 46, 12.1%); (2) Isolated post-capillary PH ((IpcPH), n = 78, 20.5%); (3) Combined pre and post-capillary PH ((CpcPH), n = 82, 21.6%). The primary endpoint was all-cause mortality at 1 year. **Results**: A total of 246 patients (64.7%) exhibited mean PAP > 20 mmHg. Overall, the presence of PH was associated with higher 1-year mortality rates (hazard ratio (HR) 2.8, 95% CI: 1.4–5.8, *p* = 0.004). Compared to patients with no PH, Pre-capPH and CpcPH (but not IpcPH) were related to higher 1-year mortality (HR 2.7, 95% CI: 1.0–7.2, *p* = 0.041 and HR 3.9, 95% CI: 1.8–8.5, *p* = 0.001, respectively). This remained significant even after the adjustment for baseline comorbidities. **Conclusions**: Pre-interventional PH according to the novel hemodynamic criteria, is linked with poor outcomes in patients undergoing TAVR for severe AS. However, this is mainly driven by patients with mean PAP ≥ 25 mmHg. Patients with a pre-capillary PH component as defined by increased PVR present an even worse prognosis as compared to patients with isolated post-capillary or no PH who present comparable 1-year mortality rates.

## 1. Introduction

Pulmonary hypertension (PH) is the hallmark of advanced-stage, severe aortic stenosis (AS) [1]. According to the traditional theory and in the absence of other significant pulmonary diseases, PH is in the clinical context, the consequence of chronic increase in pulmonary venous or post-capillary pressure, and it has been consistently associated with poor outcomes even after the replacement of the aortic valve [2,3,4].

Because of its highly significant prognostic value, right heart hemodynamics plays a key role in the baseline assessment of patients with severe AS, especially for symptomatic patients with moderate or high surgical risk, qualifying for transcatheter aortic valve replacement (TAVR) [5]. The presence of PH, especially combined pre- and post-capillary, suggests unfavorable long-term results and may, in severe cases, even put in question the clinical benefit from the valve replacement. It follows that the precise assessment of pulmonary pressure is of paramount importance.

PH was traditionally defined as the presence of a mean pulmonary artery pressure (mPAP) ≥25 mmHg, measured by right heart catheterization in the supine position and at rest [6,7]. The 6th World Symposium on Pulmonary Hypertension recently revisited the PH definition and proposed an mPAP cut-off value of 20 mmHg (Table 1) [8]. Moreover, the Task Force proposed the use of the pulmonary vascular resistance (PVR) exclusively (instead of the diastolic pressure gradient (DPG)) as a marker of the pre-capillary component of PH.

In light of the novel PH definition, it remains unclear whether the presence of pre-interventional PH preserves its prognostic significance. We, therefore, aimed to assess the effect of baseline and pre-interventional PH based on the novel definition, on clinical outcomes, in patients with AS treated with a TAVR.

## 2. Materials and Methods

### 2.1. Study Population

Data from patients who underwent a TAVR procedure in our Institution from June 2008 to December 2019 were retrospectively analyzed. The study population consisted of patients suffering from symptomatic AS of a native valve while presenting a high or intermediate risk for conventional surgical aortic valve replacement. Of 484 patients, 429 (88%) had a baseline right heart catheterization. A further 49 patients were excluded from the study for different reasons, as stated in Figure 1. The final cohort comprised 380 patients and was divided according to PAP following the revised criteria proposed by the task force at the 6th World Symposium on PH [8]. Patients with PH were further stratified in patients with pre-capillary PH ((Pre-capPH), pulmonary arterial wedge pressure (PAWP) ≤ 15 mmHg and PVR ≥ 3 Wood Units (WU), n = 46), Isolated post-capillary PH (PAWP > 15 mmHg and PVR < 3 WU, (IpcPH), n = 78) and combined pre and post-capillary ((CpcPH), PAWP > 15 mmHg and PVR ≥ 3 WU, n = 82). A group of 40 subjects presenting an mPAP > 20 mmHg, despite a PAWP and a PVR in the normal range, were considered with no PH, attributing high mPAP values to an existing hyper-dynamic state during the right heart catheterization. Data were anonymized prior to analysis. Informed written consent was obtained from each patient for inclusion in the local TAVR database as part of the Swiss prospective registry (NCT1368250) that was approved by the local Ethics Committee. A detailed study flowchart is depicted in Figure 1.

### 2.2. Invasive Hemodynamics

All patients underwent a baseline right heart catheterization as part of the standard evaluation of the AS performed either by brachial or femoral vein approach with the Seldinger technique and using a 7F Swan Ganz when possible or a 6F Arrow balloon tip catheter. Cardiac output (CO) was acquired for all patients by the thermodilution and/or the indirect Fick method. CO was also indexed to body surface area (BSA) to calculate the cardiac index (CI). Stroke volume (SV) was calculated as CO/heart rate and was also indexed to BSA (stroke volume index, SVi). Zero reference level was determined at 1/3 thoracic diameter below anterior surface [9]. A complete assessment of the pulmonary hemodynamics was performed for all patients, including the systolic, diastolic, mPAP, and PAWP. Transpulmonary pressure gradient (TPG) was calculated as the difference between the mPAP and the PAWP. DPG was also calculated as the difference between diastolic PAP and the PAWP. PVR was calculated as the ratio of TPG to CO. Pulmonary artery compliance (PAC) was defined as the ratio of the SV to the pulse PAP.

Left heart hemodynamics was also recorded including, aortic systolic, diastolic, mean, and pulse pressures, either while performing the right catheterization or on a separate day as part of the TAVR procedure and before the implantation of the aortic prosthesis. Total arterial compliance (TAC) and total vascular resistance (TVR) were also calculated in 351 patients using the simplified formulas (SV)/(Pulse pressure) and (mean arterial pressure)/(CO), respectively. For 366 patients, measures of left ventricular systolic and end-diastolic pressures during the TAVR procedure were available. Aortic valve area (AVA) was also calculated using the Gorlin formula [10]. Finally, a diagnostic coronary angiography was performed on all patients. All pressure measurements were performed using fluid recorded catheters connected to pressure transducers.

### 2.3. Baseline Echocardiographic Assessment

A complete transthoracic echocardiography was performed before the intervention on all study participants. All measurements were conducted by an experienced cardiologist with the patient in the supine position and according to standard recommendations for echocardiography [11]. Acquired images were transferred to a dedicated workstation for subsequent offline analysis (IntelliSpace Cardiovascular 5.1, Philips Medical Systems Nederland BV). Data on left ventricular geometry were collected, and left ventricular mass was calculated according to the Devereux formula. Left ventricular ejection fraction (EF) was visually estimated according to standard procedures. Parameters of left ventricular diastolic function were retrospectively collected from echocardiographic reports, including the mitral E wave maximal velocity (n = 373), the mitral A wave maximal velocity (n = 288) e’ mean (n = 370) as well as left atrial volume (n = 375), when available. Data on concomitant valvular diseases were also collected for the mitral and the tricuspid valves. Right ventricular longitudinal function was also assessed by the use of tricuspid annular plane systolic excursion (TAPSE) and pulsed Doppler peak velocity at the tricuspid annulus (DTI). AVA was calculated using the continuity equation and was also indexed to the BSA. Left atrial volume was measured for all patients according to the biplane area-length method. AS staging evaluation was also performed for all patients according to the criteria proposed by Genereux P. et al. [1].

### 2.4. Procedure Characteristics

Aortic valve replacement was performed by the implantation of the Medtronic self-expanding CoreValve and Evolut device (Medtronic Inc., Minneapolis, MN, USA, n = 342, 92%), the Edwards Sapien S3 (Edwards Lifesciences SA, Irvine, CA, USA, n = 33, 7%) or the Boston neo Accurate (Boston Scientific AG, Marlborough, MA, USA, n = 5, 1%). Device implantation success was systematically evaluated for all interventions according to the Valve Academic Research Consortium-2 consensus Document criteria [12].

### 2.5. Follow-Up

A post-TAVR follow-up was performed systematically for all patients at 1-, 6- and 12-month intervals through a clinical visit. All baseline clinical characteristics and procedural and follow-up data were stored in a dedicated database using a secured online platform (www.openclinica.com (accessed on 24 November 2020), OpenClinica LLC, Waltham, MA, USA). The primary study endpoint was all-cause mortality at one year. Events were adjudicated by an external clinical committee.

### 2.6. Statistical Analysis

The study population was divided into four groups according to the PH status: (1) Patients with no PH (NoPH), n = 174, 45.8%, reference category; (2) Pre-capPH, n = 46, 12.1%; (3) IpcPH, n = 78, 20.5% and (4) CpcPH, n = 82, 21.6%). Categorical variables are expressed as counts with percentages. Continuous variables are expressed as mean values ± standard deviation or as median and interquartile range in case of violation of the normal distribution (normality was assessed by visual inspection of histogram frequencies). Categorical variables are compared among groups by the use of Pearson Chi-square or the Fischer exact test as appropriate. For continuous variables, comparisons among groups were performed after analysis of variance (one-way ANOVA) or the Kruskal–Wallis test for variables not normally distributed. Homogeneity of variance test among groups was performed by the use of Leven’s Test, and in case of violation, Welch ANOVA was used. Pairwise comparisons between PH groups were performed by the use of post hoc tests after applying Bonferroni correction. One-year all-cause mortality rates for the four groups were calculated from Kaplan–Meier analysis (Figure 2 and Figure 3). Cox-regression analysis was performed to compute hazard ratios and the 95% confidence intervals. The proportional hazard assumption was verified for all Cox-regression models. Two multivariate Cox-regression model was used in order to adjust comparisons among groups for confounding mortality factors based on Model A: baseline EuroSCORE II and Model B: Baseline COPD, atrial fibrillation, gender, diabetes, arterial hypertension and left ventricular ejection fraction. Statistical significance was assumed at a 2-sided *p*-value level of 0.05. Statistical analysis was performed in IBM SPSS statistics (IBM Corp. Released 2020. IBM SPSS Statistics for Windows, Version 27.0. Armonk, NY, USA: IBM Corp).

## 3. Results

### 3.1. Baseline Characteristics

The baseline characteristics of the study population according to the PH status are presented in Table 2. Male gender was more frequent in the IpcPH (53%) but less frequent in the CpcPH (33%) as compared to the NoPH group (48%, *p* = 0.018). Arterial hypertension was more prevalent in the IpcPH group (92%) as compared to the NoPH group (76%, *p* = 0.029). Similarly, diabetes mellitus was more frequent in the IpcPH group (42%) as compared to the NoPH group (24%, *p* = 0.024). Patients with chronic obstructive pulmonary disease (COPD) were more prevalent in the pre-capPH group (28%) and the CpcPH group (23%), as compared to the NoPH 12%, *p* = 0.017). Atrial fibrillation/flutter was also more prevalent in all the PH groups (pre-capPH 41%; IpcPH 44%; CpcPH 48%) as compared to the NoPH group (20%, *p* < 0.001). Accordingly, oral anticoagulation and beta-blockers intake was more frequent in the IpcPH (39% and 45%) and the CpcPH groups (49% and 55%) as compared to the NoPH group (20% and 31%, *p* < 0.001 and *p* = 0.002 accordingly). Patients with CpcPH were more symptomatic (NYHA stage III or IV 84%) as compared to the NoPH group (67%, *p* = 0.018). Finally higher pre-interventional risk scores were noted in the PH groups as compared to the NoPH group both in terms of EuroSCORE II (CpcPH, 24.6% [16.1–39.9] vs. IpcPH, 14.9% [11.4–23.5] vs. Pre-capPH, 13.1% [9.7–20.1] vs. NoPH, 12.1% [8.4–16.7], *p* < 0.001) as well as Society for Thoracic Surgeons (STS) Score (CpcPH, 6.7% [3.9–9.8] vs. IpcPH, 5.0% [3.6–9.1] vs. Pre-capPH, 5.3% [3.7–8.0] vs. NoPH, 4.1% [2.9–6.4], *p* < 0.001).

### 3.2. 2015. ESC Guidelines vs. 6th World Symposium PH Definition Groups

PH group classification according to each time definition used is presented in Figure 2. A total of 14 patients (3.7%) were classified from the NoPH to the Pre-capPH group, 9 patients (2.4%) from the NoPH to the IpcPH group, and 12 (3.2%) patients from the Pre-capPH to the NoPH group. No significant difference in groups was noted between the two definitions (*p* = 0.818).

### 3.3. Echocardiographic and Heart Catheterization Parameters

Echocardiographic and invasive hemodynamic characteristics of the study population according to the PH groups are presented in Table 3 and Table 4. Baseline AVA was smaller in the CpcPH group as assessed by both the continuity equation (0.65 ± 0.2 cm^2^) and the Gorlin formula (0.46 ± 0.14 cm^2^), compared to the NoPH group (0.77 ± 0.2 cm^2^ and 0.60 ± 0.28 cm^2^, respectively, *p* < 0.001 for both). Left ventricular end-diastolic diameter was higher in the IpcPH (4.8 ± 0.9 cm) and the CpcPH group (4.8 ± 0.7 cm) as compared to the NoPH group (4.4 ± 0.7 cm, *p* = 0.002 and *p* = 0.003, accordingly). Left ventricular EF was also lower in the IpcPH group (58 [43–65]%) and the CpcPH group (55 [41–63]%) as compared to the NoPH group (63 [59–65]%, p < 0.001). Mitral E wave maximal velocity was higher in the IpcPH (109 ± 34 cm/s, p < 0.001) and the CpcPH group (116 ± 37 cm/s, *p* < 0.001) as compared to the NoPH group (84 ± 31 cm/s, p < 0.001). In accordance, the IpcPH and CpcPH groups presented lower mitral A wave maximal velocity values (92 ± 32 cm/s and 83 ± 37 cm/s, p = 0.006 and p < 0.001, accordingly), compared to the NoPH group (110 ± 33 cm/s, p < 0.001). Mean e’ was comparable among the groups. Left atrial volume was higher in the IpcPH (87 [70–108] mL) and the CpcPH (83 [74–107] mL) as compared to the NoPH group (71 [58–85] mL, p < 0.001). TAPSE was lower in the CpcPH group (17 ± 5.1 mm) as compared to the NoPH group (21 ± 4.5 mm, p < 0.001). Similarly, DTI was lower in the CpcPH group (10.2 ± 2.9 cm/s) compared to the NoPH group (11.9 ± 2.7 cm/s, *p* < 0.001).

Both pre-capPH and CpcPH exhibited lower SV, SVi, CO, and CI as compared to the NoPH group (Table 4, *p* < 0.05 for all), while the CpcPH group presented higher heart rate (vs. NoPH, *p* = 0.003). Aortic systolic, diastolic, and mean pressures were comparable among groups. Patients in the CpcPH group exhibited lower TAC and higher TVR than the NoPH group (*p* < 0.05 for all). Finally, the CpcPH group exhibited the highest PAP (systolic, diastolic, and mean), decreased PAC, and high PVR and Ea (*p* < 0.05 for all, Table 4).

### 3.4. TAVR Intervention

Data on the TAVR procedure are presented in Table 5. Femoral access was the preferred approach for most patients (95%), followed by trans-apical (2%) and subclavian access (2%). Forty-seven patients (12.4%) underwent a concomitant procedure (coronary angioplasty). Device success was achieved in 346 interventions (91%), which was comparable among the groups.

### 3.5. Clinical Outcomes

Clinical follow-up was completed for the totality of the study population. Data on 1-year all-cause mortality as well unadjusted and adjusted hazard ratios are presented in Table 6. Figure 3 presents mortality data stratified according to the PH status, mPAP quartiles, the presence of mean PAP 21–24 mmhg, and type of PH (Kaplan–Meier analysis). Compared to patients with No PH, patients with PH exhibited a higher overall mortality rate at 1 year (15.5% vs. 5.7%, unadjusted HR 2.8; 95% CI: 1.4–5.8, *p* = 0.004). This remained significant even after adjustment for baseline EuroSCORE II (Model A, adjusted HR 2.5; 95% CI: 1.2–5.3, *p* = 0.013) and baseline comorbidities (Model B, adjusted HR 2.7; 95% CI: 1.3–5.7, *p* = 0.011). Compared to the NoPH group, patients with Pre-capPH presented higher all-cause mortality rates at 1 year (15.2% vs. 5.7%, unadjusted HR 2.7; 95% CI: 1.0–7.0, *p* = 0.041). The association remained significant after adjustment for baseline EuroSCORE II (Model A, adjusted HR 2.8; 95% CI: 1.1–7.4, *p* = 0.037) and baseline comorbidities (Model B, adjusted HR 2.7; 95% CI: 1.0–7.4, *p* = 0.049). Similarly, compared to the NoPH group, patients with CpcPH presented higher all-cause mortality rates at 1 year (20.7% vs. 5.7%, unadjusted HR 3.9; 95% CI: 1.8–8.5, *p* = 0.001). The association remained significant after adjustment for baseline EuroSCORE II (Model A, adjusted HR 3.7; 95% CI: 1.6–8.6, *p* = 0.003) and baseline comorbidities (Model B, adjusted HR 3.9, 95% CI: 1.7–9.1, *p* = 0.001). Patients with mean PAP between 21 and 24 mmHg presented comparable 1-year mortality rates as compared to patients with mean PAP ≤ 20 mmHg (8.7% vs. 6.4%, unadjusted HR: 0.87, 95% CI: 0.23–3.2, *p* = 0.83). The exclusion from the analysis of patients with a “hyperdynamic” PH (mean PAP > 20 mmHg, PVR < 3 WU and PAWP > 15 mmHg, did not alter the findings (unadjusted HR 1.3, 95% CI: 0.28–5.97, *p* = 0.744). Figure 3 presents 1-year all-cause mortality rates when the population was stratified according to the PAWP, PVR, PAC, and AS staging. Patients with PAWP exceeding 15 mmHg at baseline presented higher all-cause mortality rates at 1 year (15.8% vs. 7.4%, unadjusted HR 2.22; 95% CI: 1.19–4.15, *p* = 0.012) as compared to patients with PAWP lower or equal to 15 mmHg. Accordingly, patients with high PVR (≥3 WU) exhibited higher mortality rates at 1 year (16.6% vs. 7.4%, unadjusted HR 2.32; 95% CI: 1.25–4.29, *p* = 0.007) as compared to patients with low PVR. Moreover, patients with low PAC (infra-median) presented higher mortality rates as compared to patients with compliant (supra-median) pulmonary artery (14.7% vs. 7.4%, unadjusted HR 2.07; 95% CI: 1.09–3.94, *p* = 0.026). Finally, patients with AS staging > 2 (right chamber involvement) presented higher 1-year mortality rates as compared to patients with AS staging ≤ 2 (12.8% vs. 4.8%, unadjusted HR 2.82, 95% CI: 1.00–7.9, *p* = 0.048).

## 4. Discussion

The main findings of the present study can be summarized as follows: (1) In patients with severe AS undergoing TAVR, the presence of pre-interventional PH, according to the novel proposed hemodynamic criteria, is associated with higher 1-year mortality rates even after adjustment for baseline comorbidities. (2) This is mainly driven by patients exhibiting mean PAP equal to or higher than 25 mmHg, (3) worse prognosis is observed in patients with a significant pre-capillary component (Pre-capPH or CpcPH, defined as a PVR ≥ 3 WU), as compared to patients with pure post-capillary or no PH.

Our findings are in accordance with previous reports on PH and outcomes in patients with AS after TAVR and suggest that despite the novel definition, PH preserves its prognostic significance [3,4,13,14,15]. Moreover, in a recent study by Maeder et al., despite a slight reclassification effect, PH remained a significant determinant of all-cause mortality. At the same time, the high PVR criterion displayed the strongest association with poor outcomes [16].

Independently of the responsible underlying mechanism, data on the prognostic significance of previously called “borderline” mPAP (21–24 mmHg) are limited. In a 2017 report, Douschan and coworkers observed that an mPAP between 21 and 24 mmHg was associated with poor survival, as compared to patients with a “lower normal” mPAP, even after adjustment for baseline comorbidities, in a series of 547 patients with unexplained dyspnea with a median follow-up time of 45.9 months [17]. Moreover, in patients with systemic sclerosis, an mPAP of 21–24 mmHg was associated with the development of overt PH (mPAP ≥ 25 mmHg) and a 3-year mortality rate of 18% [18]. In another study of 21,727 patients referred for cardiac catheterization, the adjusted risk of mortality increased significantly for an mPAP of ≥ 19 mmHg, while patients with mPAP between 19 and 24 mmHg exhibited a 23% higher all-cause mortality as compared to the “lower normal” group [19]. In accordance, the Vanderbilt University Cohort showed that in a gender-balanced referral population, the presence of “borderline” mPAP was associated with a 31% increase in adjusted all-cause mortality [20]. Moreover, Lau and coworkers demonstrated that resting mPAP of 21–24 mmHg is a strong predictor of severe hemodynamic impairment during effort with the development of exercise PH [21]. These data, collectively, and a landmark meta-analysis reviewing data from 1200 healthy controls showing that the normal upper limit rarely exceeds 20 mmHg lead to the recently proposed revision of the PH definition [20]. Interestingly, our study failed to demonstrate a poor prognosis for patients with a pre-interventional mPAP between 21 and 24 mmHg, as compared to subjects with mPAP ≤ 20 mmHg (Figure 3). This finding should, however, be interpreted with caution as our study was not designed to specifically address this question (lack of statistical power). Moreover, it remains unclear whether this lack of association persists during longer follow-up periods.

Regardless of the cut-off value used, our results confirm the prognostic significance of PH in the setting of a severe AS treated with a TAVR, underlying the role of PH as a marker of advanced-stage severe AS. In the study of Genereux et al., the presence of PH or tricuspid valve damage (stage 3 AS) was independently associated with increased mortality after TAVR [1]. The proposed anatomic and functional cardiac staging system was the strongest predictor of mortality in a cohort of 1661 patients who underwent a TAVR, with a 1-year mortality risk increase of 45% for each stage increment. Contrary to the traditional hypothesis, the authors observed the extent of the cardiac damage reflected by the proposed staging system, which did not seem to occur in a sequential fashion. This suggests that cardiac damage related to left ventricular overload may vary according to the patient’s susceptibility and possibly genetic characteristics [1].

In the context of severe AS, IpcPH is the most common form of PH [22]. The underlying mechanism involves a progressive exposure of the pulmonary circulation to high pressures exclusively due to the increasing filling pressures of the left ventricle and is generally considered reversible. This is further supported by previous findings showing a decrease in residual pulmonary systolic pressure in patients with IpcPH or CpcPH but not in patients with pre-capPH [4]. This reversibility may explain the lack of association with 1-year mortality seen in the IpcPH group. Our results are consistent with the study of O’Sullivan and coworkers, where patients were stratified according to the traditional PH definition [4].

The presence of a pre-capillary PH component as assessed by the novel PVR criterion (>20 mmHg and PVR ≥ 3 WU) was observed in a large part of the study population (Pre-capPH and CpcPH, n = 128, 33.6%). The CpcPH group exhibited the worse outcomes with a ~4-fold increase in 1-year all-cause mortality compared to patients with no PH even after adjustment for differences in baseline comorbidities. Female gender was more prevalent in this group, as well as COPD and atrial fibrillation, while patients in this group presented with more severe respiratory symptoms. Moreover, the CpcPH group exhibited lower left ventricular EF, lower right ventricular longitudinal function and more advanced AS staging. In terms of left-sided hemodynamics, the CpcPH group was characterized by decreased SV and CO as well as stiffer arterial trees and higher vascular resistance. In terms of the underlying mechanism, the presence of a pre-capillary PH component may be the reflection of a long-standing, passive exposure of the pulmonary circulation to high left ventricular filling pressures, finally leading to pulmonary vascular remodeling and increased vascular resistance [23].

A total of 46 patients (12.1%) presented with pure pre-capillary PH. Although this may seem counter-intuitive, pre-capillary PH has been consistently observed in patients with AS undergoing TAVR [4,15]. This group exhibited worse outcomes as compared to patients with NoPH. Possible explanations of isolated pre-capillary PH in patients with AS include the presence of concomitant pulmonary disease (e.g., COPD) or altered hemodynamic conditions during the right heart catheterization (e.g., excessive use of diuretics) inducing lower (or even normal) left ventricular filling pressures. However, the concomitant pulmonary disease cannot explain the observed association with poor outcomes since the association with 1-year mortality remained significant even after the adjustment for baseline comorbidities.

## 5. Limitations

The study is subjective to the limitations of a retrospective, single-center, cohort study design, with prospectively collected data. Moreover, CO was acquired invasively by two different techniques (thermodilution or the indirect Fick method with estimated oxygen consumption), which may not be used interchangeably. Moreover, only patients who underwent a baseline right heart catheterization were included in the study, and thus this is not a consecutive patient series. Thus, results may not be directly extrapolated to all patients undergoing TAVR for severe symptomatic AS. In addition, echocardiographic data on diastolic dysfunction were exported from routine echocardiography reports, and the EF was visually estimated. Moreover, the classification of subjects with mPAP > 20 mmHg, but PAWP ≤ 15 mmHg and PVR < 3 WU in the No PH group, may be subject to discussion as this category is not included in the recently proposed definition of PH. Finally, the present study was not designed to address the prognostic significance of patients with mean PAP between 21 and 24 mmHg, and thus results for this group should be interpreted with caution.

## 6. Conclusions

Pre-interventional PH based on the novel hemodynamic criteria is still associated with poor outcomes in patients undergoing a TAVR for severe AS. This is mainly driven by patients exhibiting mean PAP equal to or higher than 25 mmHg. Patients with a pre-capillary PH component are characterized by extensive cardiac damage, which is associated with a worse prognosis. Careful hemodynamic evaluation prior to and after TAVR may help physicians in better defining procedural outcomes but also patients’ responses to treatment.

## Figures and Tables

**Figure 1 jcm-11-04279-f001:**
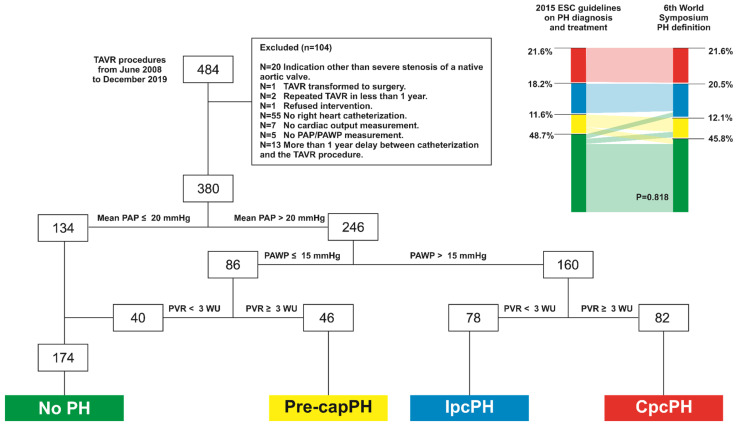
Study flow chart, and PH groups of the study population, according to the novel definition. Reclassification effect of PH groups as compared to the criteria proposed in the 2015 ESC guidelines on PH diagnosis and treatment. TAVR: Transcatheter aortic valve replacement; PA: Pulmonary artery pressure; PAWP: pulmonary arterial wedge pressure; PVR: pulmonary vascular resistance; WU: Wood Units; PH: pulmonary hypertension; Pre-cap: Pre-capillary; Ipc: Isolated post-capillary; Cpc: Combined pre and post-capillary.

**Figure 2 jcm-11-04279-f002:**
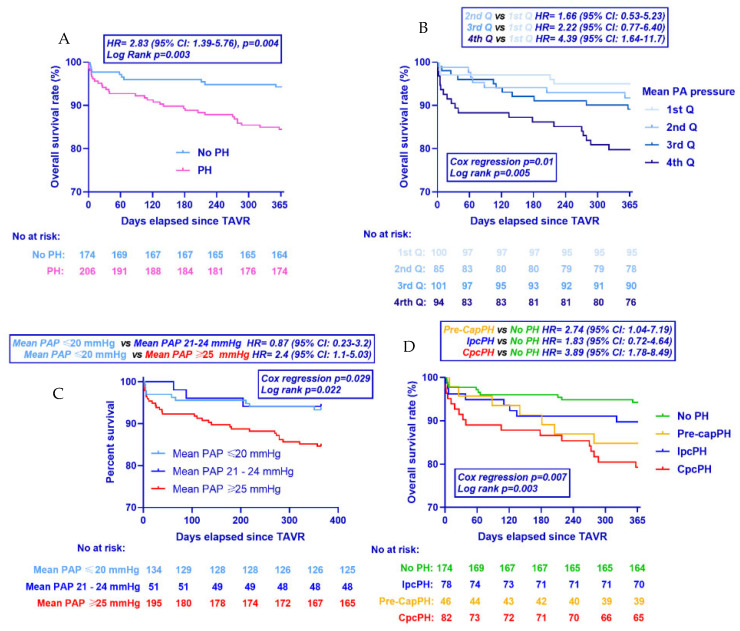
Kaplan–Meier curves for all-cause mortality at 1 year. Population stratified according to (**A**) whether PH was present or not, (**B**) mPAP quartiles, (**C**) the presence of mean PAP between 21 and 24 mmHg and (**D**) hemodynamic type of PH. HR: hazard ratio; *p* values refer to unadjusted hazard ratios from Cox regression analysis. CI: confidence intervals; TAVR: Transcatheter Aortic Valve Replacement. PH: Pulmonary hypertension; AS: Aortic stenosis; PA; Pulmonary artery.

**Figure 3 jcm-11-04279-f003:**
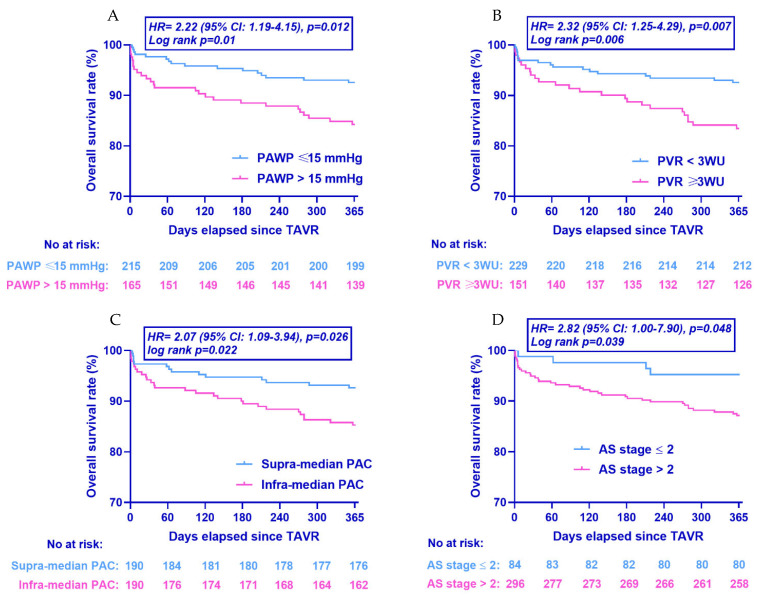
Kaplan–Meier curves for all-cause mortality at 1 year. Population stratified according to (**A**) PAWP, (**B**) PVR, (**C**) PAC, and (**D**) AS staging. HR: hazard ratio; *p* values refer to unadjusted hazard ratios from Cox regression analysis. CI: confidence intervals; TAVR: Transcatheter Aortic Valve Replacement; PAWP: Pulmonary artery wedge pressure; PVR: Pulmonary vascular resistance; PAC: Pulmonary artery compliance; AS: Aortic stenosis.

**Table 1 jcm-11-04279-t001:** PH hemodynamic criteria and definitions according to the Proceedings of the 6th World Symposium and the 2015 ESC Guidelines on the diagnosis and treatment of pulmonary hypertension.

Definition	6th World Symposium Criteria	2015 ESC Guidelines For PH Diagnosis	Pathophysiological Mechanisms
**No PH**	Mean PAP ≤ 20 mmHg *	Mean PAP < 25 mmHg	
**Pre-capillary PH** **(Pre-capPH)**	Mean PAP > 20 mmHg and PAWP ≤ 15 mmHg and PVR ≥ 3 WU	Mean PAP ≥ 25 mmHg and PAWP ≤ 15 mmHg	1. Pulmonary arterial hypertension 3. PH due to lung diseases or hypoxia 4. PH due to pulmonary artery obstructions 5. PH with unclear and/or multifactorial mechanisms
**Isolated post-capillary PH (IpcPH)**	Mean PAP > 20 mmHg and PAWP > 15 mmHg and PVR < 3 WU	Mean PAP ≥ 25 mmHg and PAWP > 15 mmHg and DPG < 7 mmHg and/or PVR ≤ 3WU ^†^	2. PH due to left heart disease - Heart failure with preserved ejection fraction - Heart failure with reduced ejection fraction - Valvular heart disease - Congenital/acquired conditions leading to post-capillary PH 5. PH with unclear and/or multifactorial mechanisms
**Combined pre-and post-capillary PH (CpcPH)**	Mean PAP > 20 mmHg and PAWP > 15 mmHg and PVR ≥ 3 WU	Mean PAP ≥ 25 mmHg and PAWP > 15 mmHg and DPG ≥ 7 mmHg and/or PVR > 3 WU ^†^	

* Unless existing hyper-dynamic state defined as mPAP > 20 mmHg with PAWP ≤ 15 mmHg and PVR < 3WU). ^†^ In case of discrepancy between DPG and PVR, the PVR criterion was used. PH: Pulmonary hypertension; PAP: Pulmonary artery pressure; PAWP: pulmonary arterial wedge pressure; PVR: pulmonary vascular resistance; WU: Wood Units.

**Table 2 jcm-11-04279-t002:** Baseline characteristics of the study population according to the PH groups.

		PH			
	NoPH	Pre-capPH	IpcPH	CpcPH	*p*
	n = 174	n = 46	n = 78	n = 82	Value
**Demographics**					
Age (years)	84 ± 6	84 ± 6	82 ± 7	84 ± 6	0.165
Height (cm)	165 ± 9	163 ± 9	167 ± 9	163 ± 9	0.014
Weight (kg)	72 ± 15	68 ± 11	74 ± 15	69 ± 15	0.058
BMI (kg/m^2^)	26.3 ± 4.8	25.7 ± 4.2	26.5 ± 4.6	25.9 ± 5.7	0.767
BSA (m^2^)	1.81 ± 0.21	1.75 ± 0.18	1.84 ± 0.22	1.75 ± 0.21	0.021
Gender (males, n, %)	84 (48)	15 (33)	41 (53) *‡*	27 (33) *‡*	0.018
**Pre-intervention risk scores**					
EuroSCORE (%, n = 372)	12.1 [8.4–16.7]	13.1 [9.7–20.1]	14.9 [11.4–23.5] *‡*	24.6 [16.1–39.9] *‡*	<0.001
STS Score (%, n = 372)	4.1 [2.9–6.4]	5.3 [3.7–8.0] *‡*	5.0 [3.6–9.1] *‡*	6.7 [3.9–9.8] *‡*	<0.001
**Comorbidities and risk factors**					
Diabetes (n, %)	41 (24)	13 (28)	33 (42) *‡*	22 (27)	0.024
Dyslipidaemia (n, %)	119 (68)	32 (70)	55 (71)	57 (70)	0.989
Arterial hypertension (n, %)	133 (76)	36 (78)	72 (92) *‡*	65 (79)	0.029
Smokers (n, %)	9 (5)	4 (9)	8 (10)	7 (9)	0.482
CAD (n, %)	93 (53)	21 (46)	45 (58)	48 (59)	0.496
Previous MI (n, %)	22 (13)	5 (11)	9 (12)	14 (17)	0.675
PAD (n, %)	16 (9)	8 (17)	14 (18)	17 (21)	0.055
COPD (n, %)	20 (12)	13 (28) *‡*	16 (21)	19 (23) *‡*	0.017
Renal failure (n, %)	83 (48)	20 (43)	45 (58)	45 (55)	0.298
Cancer (n, %)	34 (20)	12 (26)	15 (19)	17 (21)	0.787
Atrial fibrillation/flutter (n, %)	35 (20)	19 (41) *‡*	34 (44) *‡*	39 (48) *‡*	<0.001
**Presence of symptoms**					
NYHA III or IV (n, %)	116 (67)	37 (80)	57 (73)	69 (84) *‡*	0.018
Syncope (n, %, n = 368)	29 (17)	3 (7)	5 (7)	7 (9)	0.051
Angina (n, %, n = 368)	33 (19)	8 (18)	14 (19)	18 (23)	0.894
**Baseline medications**					
Aspirin (n, %)	98 (56)	25 (54)	43 (55)	43 (52)	0.951
Oral anticoagulation (n, %)	34 (20)	15 (33)	30 (39) *‡*	40 (49) *‡*	<0.001
Beta-blockers (n, %)	53 (31)	20 (44)	35 (45) *‡*	45 (55) *‡*	0.002
ACE inhibitors (n, %)	39 (22)	11 (24)	20 (26)	17 (21)	0.896
ARBs (n, %)	60 (35)	13 (28)	34 (44)	25 (31)	0.240
Ca channel blockers (n, %)	28 (16)	9 (20)	22 (28)	19 (23)	0.151
Statin (n, %)	101 (58)	26 (57)	44 (56)	42 (51)	0.785

BMI: Body mass index; BSA: Body surface area; STS: Society of Thoracic Surgeons; CAD: Coronary artery disease; PAD: Peripheral artery disease; COPD: Chronic obstructive pulmonary disease; NYHA: New York Heart Association; ACE: Angiotensin-converting enzyme inhibitors; ARBs; Angiotensin receptor blockers. *‡* vs. NoPH *p* < 0.05.

**Table 3 jcm-11-04279-t003:** Echocardiographic parameters according to the PH groups.

		PH			
	NoPH	Pre-capPH	IpcPH	CpcPH	*p*
	n = 174	n = 46	n = 78	n = 82	Value
**Aortic valve stenosis severity**					
Transvalvular mean pressure gradient (mmHg, n = 378)	43 ± 13	42 ± 17	40 ± 14	41 ± 14	0.051
Transvalvular max pressure gradient (mmHg, n = 378)	73 ± 21	70 ± 25	68 ± 22	65 ± 20 *‡*	0.044
Transvalvular max velocity (cm/s, n = 378)	422 ± 62	413 ± 75	408 ± 66	398 ± 61 *‡*	0.046
AVA (cm^2^, n = 378)	0.77 ± 0.2	0.69 ± 0.19	0.73 ± 0.21	0.65 ± 0.2 *‡*	<0.001
AVA indexed for BSA (cm^2^/m^2^, n = 378)	0.43 ± 0.12	0.40 ± 0.11	0.40 ± 0.12	0.37 ± 0.11 *‡*	0.002
**LV geometry**					
LV End-diastolic diameter (cm, n = 377)	4.4 ± 0.7	4.5 ± 0.7	4.8 ± 0.9 *‡*	4.8 ± 0.7 *‡*	<0.001
LV mass (g, n = 375)	199 ± 69	201 ± 73 *‡*	228 ± 68 *‡*	208 ± 62	0.019
LV mass indexed for BSA (g/m^2^, n = 375)	110 ± 33	115 ± 41	120 ± 35 *‡*	116 ± 36 *‡*	0.009
RWT (n = 374)	0.48 ± 0.13	0.47 ± 0.11	0.45 ± 0.14	0.44 ± 0.13	0.082
**LV systolic function**					
Ejection fraction (%, n = 378)	63 [59–65]	63 [55–66]	58 [43–65] *‡*	55 [41–63] *‡*	<0.001
LVOT flow max (mL/s, n = 351)	254 ± 80	201 ± 60 *‡*	268 ± 90	197 ± 66 *‡*	<0.001
**LV diastolic function**					
Mitral E wave maximal velocity (cm/s, n = 373)	84 ± 31	69 ± 31	109 ± 34 *‡*	116 ± 37 *‡*	<0.001
Mitral A wave maximal velocity (cm/s, n = 288)	110 ± 33	116 ± 33	92 ± 32 *‡*	83 ± 37 *‡*	<0.001
e’ mean (m/s, n = 370)	5.4 ± 1.7	5.5 ± 1.7	5.8 ± 2.2	5.5 ± 1.8	0.450
Left atrial volume (mL, n = 375)	71 [58–85]	76 [65–87]	87 [70–108] *‡*	83 [74–107] *‡*	<0.001
Left atrial volume indexed BSA (mL/m^2^, n = 375)	40 [32–48]	43 [34–51]	46 [36–62] *‡*	49 [41–60] *‡*	<0.001
**RV longitudinal function**					
TAPSE (mm, n = 372)	21 ± 4.5	20 ± 4.5	20 ± 5.0	17 ± 5.1 *‡*	<0.001
DTI (cm/s, n = 370)	11.9 ± 2.7	11.6 ± 2.6	11.3 ± 3.0	10.2 ± 2.9 *‡*	<0.001
**Aortic regurgitation**					0.866
None (%)	42 (24)	46 (17)	78 (31)	82 (28)	
Discrete (%)	117 (67)	32 (69)	48 (61)	52 (63)	
Discrete to moderate (%)	7 (4)	3 (7)	4 (5)	3 (4)	
Moderate (%)	8 (5)	3 (7)	2 (3)	4 (5)	
**Mitral regurgitation**			*‡*	*‡*	<0.001
None (%)	87 (50)	18 (39)	25 (32)	24 (29)	
Discrete (%)	75 (43)	21 (46)	41 (53)	35 (43)	
Discrete to moderate (%)	9 (5)	6 (13)	7 (9)	11 (13)	
Moderate (%)	3 (2)	1 (2)	5 (6)	12 (15)	
**Tricuspid regurgitation**		*‡*	*‡*	*‡*	<0.001
None (%)	127 (73)	23 (50) *‡*	43 (55)	27 (33)	
Discrete (%)	41 (23)	15 (33)	23 (30)	34 (42)	
Discrete to moderate (%)	3 (2)	5 (11) *‡*	4 (5)	11 (13)	
Moderate (%)	3 (2)	2 (4)	7 (9)	7 (9)	
Moderate to severe (%)	0 (0)	1 (2)	1 (1)	3 (4)	

LV: Left ventricle; AVA: Aortic valve area; RWT: Relative wall thickness; RV: Right ventricle; TAPSE: Tricuspid annular plane systolic excursion; DTI: Pulse Doppler peak velocity at the tricuspid annulus; BSA: Body surface area. *‡* vs. No PH *p* < 0.05.

**Table 4 jcm-11-04279-t004:** Right and left heart catheterization parameters according to the PH groups.

		PH			
	No PH	Pre-capPH	IpcPH	CpcPH	*p*
	n = 174	n = 46	n = 78	n = 82	Value
**Aortic valve stenosis severity**					
Transvalvular mean pressure gradient (mmHg, n = 343)	33 ± 13	36 ± 18	32 ± 16	31 ± 14	0.470
Transvalvular peak to peak pressure gradient (mmHg, n = 366)	44 ± 18	49 ± 24	42 ± 23	41 ± 23	0.312
Transvalvular pressure gradient AUC (mmHg·s, n = 343)	14 ± 6	15 ± 9	13 ± 6	13 ± 6	0.126
AVA (Gorlin, cm^2^, n = 341)	0.60 ± 0.28	0.51 ± 0.23	0.66 ± 0.25	0.46 ± 0.14 *‡*	<0.001
AVA indexed for BSA (Gorlin, cm^2^/m^2^, n = 341)	0.33 ± 0.14	0.29 ± 0.13	0.36 ± 0.14	0.26 ± 0.08 *‡*	<0.001
**Aortic valve stenosis staging**		*‡*	*‡*	*‡*	<0.001
Stage 0	10 (0)	0 (0)	0 (0)	0 (0)	
Stage 1	10 (6)	2 (4)	1 (1)	0 (0)	
Stage 2	55 (34)	1 (2)	5 (6)	0 (0)	
Stage 3	4 (2)	4 (9)	14 (18)	1 (1)	
Stage 4	95 (55)	39 (85)	58 (74)	81 (99)	
**Systemic afterload**					
Aortic systolic blood pressure (mmHg)	128 ± 25	120 ± 28	119 ± 27	126 ± 27	0.070
Aortic diastolic blood pressure (mmHg)	54 ± 11	52 ± 13	53 ± 12	56 ± 15	0.192
Aortic pulse pressure (mmHg)	74 ± 20	68 ± 24	66 ± 24 *‡*	69 ± 24	0.051
Aortic mean pressure (mmHg)	82 ± 15	78 ± 17	79 ± 16	83 ± 17	0.112
Total arterial compliance (mL/mmHg, n = 351)	0.58 ± 0.27	0.51 ± 0.26	0.67 ± 0.33	0.46 ± 0.2 *‡*	<0.001
Total vascular resistance (mmHg·min/mL, n = 351)	1.38 ± 0.45	1.54 ± 0.47	1.27 ± 0.45	1.82 ± 0.73 *‡*	<0.001
Zva (mmHg/mL/m^2^, n = 345)	5.49 ± 1.6	6.46 ± 1.68 *‡*	5.24 ± 1.60	7.23 ± 2.46 *‡*	<0.001
**LV systolic function**					
Stroke volume (mL)	60 ± 17	48 ± 15 *‡*	59 ± 17	43 ± 15 *‡*	<0.001
Stroke volume index (mL/m^2^)	33 ± 7	27 ± 7 *‡*	32 ± 8	25 ± 8 *‡*	<0.001
Cardiac output (L/min)	4.2 ± 1.1	3.7 ± 0.9 *‡*	4.4 ± 1.1	3.3 ± 0.8 *‡*	<0.001
Cardiac Index (L/min/m^2^)	2.3 ± 0.5	2.1 ± 0.4 *‡*	2.4 ± 0.5	1.9 ± 0.4 *‡*	<0.001
Heart rate (bpm)	65 ± 10	69 ± 14	68 ± 14	71 ± 17 *‡*	0.002
LV systolic pressure (mmHg, n = 366)	172 ± 28	169 ± 37	161 ± 29	165 ± 31	0.055
LV end-diastolic pressure (mmHg, n = 366)	15 ± 7	18 ± 10	18 ± 7 *‡*	20 ± 8 *‡*	<0.001
**Right heart hemodynamics**					
Pulmonary arterial wedge pressure (mmHg)	9 ± 4	12 ± 3 *‡*	23 ± 6 *‡*	23 ± 5 *‡*	<0.001
Pulmonary arterial systolic pressure (mmHg)	33 ± 7	49 ± 12 *‡*	49 ± 10 *‡*	64 ± 13 *‡*	<0.001
Pulmonary arterial diastolic pressure (mmHg)	10 ± 4	15 ± 5 *‡*	19 ± 6 *‡*	24 ± 7 *‡*	<0.001
Pulmonary arterial mean pressure (mmHg)	18 ± 4	28 ± 5 *‡*	30 ± 5 *‡*	39 ± 8 *‡*	<0.001
Pulmonary arterial compliance (mL/mmHg)	2.8 ± 1.1	1.5 ± 0.5 *‡*	2.0 ± 0.7 *‡*	1.2 ± 0.5 *‡*	<0.001
Transpulmonic gradient (mmHg)	9 ± 3	16 ± 5 *‡*	8 ± 3	17 ± 6 *‡*	<0.001
Pulmonary diastolic pressure gradient (mmHg)	1 ± 4	3 ± 4 *‡*	−4 ± 5 *‡*	1 ± 6	<0.001
Pulmonary vascular resistance (mmHg·min/mL)	2.1 ± 0.8	4.4 ± 1.5 *‡*	1.8 ± 0.7	5.3 ± 2.5 *‡*	<0.001
Effective pulmonary arterial elastance (mmHg/mL)	0.38 ± 0.19	1.1 ± 0.38 *‡*	0.91 ± 0.36 *‡*	1.61 ± 0.65 *‡*	<0.001

LV: Left ventricle; AVA: Aortic valve area; RWT: Relative wall thickness; RV: Right ventricle; TAPSE: Tricuspid annular plane systolic excursion; DTI: Pulse Doppler peak velocity at the tricuspid annulus; BSA: Body surface area. *‡* vs. No PH *p* < 0.05.

**Table 5 jcm-11-04279-t005:** TAVR procedural characteristics.

		PH			
	No PH	Pre-capPH	IpcPH	CpcPH	*p*
	n = 174	n = 46	n = 78	n = 82	Value
**Access site**					0.086
Femoral (n, %)	170 (98)	42 (91)	75 (96)	74 (90)	
Apical (n, %)	1 (1)	2 (5)	1 (1)	4 (4)	
Sub-clavian (n, %)	3 (1)	1 (2)	1 (1)	2 (2)	
Other (n, %)	0 (0)	1 (2)	1 (1)	3 (4)	
**Prosthetic valve type**					0.962
Medtronic CoreValve (n, %)	156 (90)	43 (94)	68 (87)	75 (92)	
Edwards Sapien (n, %)	15 (8)	3 (7)	9 (12)	6 (7)	
Boston Acurate (n, %)	3 (2)	0 (0)	1 (1)	1 (1)	
**Procedural specifications**					
Concomitant procedure (n, %)	19 (11)	6 (13)	7 (9)	15 (18)	0.279
Device success (n, %)	162 (93)	41 (89)	73 (94)	70 (85)	0.179

**Table 6 jcm-11-04279-t006:** Unadjusted and adjusted all-cause mortality rates at 1 year.

		PH			PH vs. No PH		Pre-capPH vs. No PH		IpcPH vs. No PH		CpcPH vs. No PH	
	No PH	Pre-capPH	IpcPH	CpcPH								
	n = 174	n = 46	n = 78	n = 82	HR (95% CI)	*p* Value	HR (95% CI)	*p* Value	HR (95% CI)	*p* Value	HR (95% CI)	*p* Value
**Unadjusted**												
All-cause death (n, %)	10 (5.7)	7 (15.2)	8 (10.3)	17 (20.7)	2.8 (1.4–5.8)	0.004	2.7 (1.0–7.2)	0.041	1.8 (0.7–4.6)	0.202	3.9 (1.8–8.5)	0.001
**Adjusted Model A**												
All-cause death (n, %)					2.5 (1.2–5.3)	0.013	2.8 (1.1–7.4)	0.037	1.6 (0.6–8.6)	0.361	3.7 (1.6–8.6)	0.003
**Adjusted Model B**												
All-cause death (n, %)					2.7 (1.3–5.7)	0.011	2.7 (1.0–7.4)	0.049	1.8 (0.7–4.8)	0.248	3.9 (1.7–9.1)	0.001

Adjusted HRs [CI] are obtained from multivariable Cox regression analysis with Model A covariates: baseline EuroScore II; Model B covariates: baseline COPD, atrial fibrillation, left ventricular ejection fraction, gender, diabetes and arterial hypertension.

## Data Availability

The datasets presented in this article are not readily available because of institutional restrictions applying to data involving human subjects. Requests to access the datasets should be directed to D.A., dionysios.adamopoulos@hcuge.ch and S.N., stephane.noble@hcuge.ch.

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
