# Peer review of "Prognostic Implications of the Novel Pulmonary Hypertension Definition in Patients with Aortic Stenosis after Transcatheter Valve Replacement"

_jcm, 2022, doi:10.3390/jcm11154279_

Round 1

Reviewer 1 Report

Thank you for letting me review the article entitled ‘Prognostic implication of the Novel Pulmonary Hypertension in Patients with Aortic Stenosis after Transcatheter Valve Replacement’. The authors did a meticulous work to verify the adverse outcome predictability of updated pulmonary hypertension (PH) criteria for 380 patients with transcatheter aortic valve replacement (TAVR). The outcome variable was all-cause mortality during the first post-procedural year, and decreased time to death was analyzed by Cox regression. Based on the newly defined PH classification, they broke down the cohort into four groups for subgroup analysis: no PH group (n=174), pre-capillary PH (Pre-capPH, n=46), isolated post-capillary PH (IpcPH, n=78), and combined pre- and post-capillary PH (CpcPH, n=82). They found that 1) the association of pre-TAVR PH (i.e. mPAP > 20 mmHg by new definition) was a significant risk factor for decreased time to death, and 2) survival of IpcPH group was comparable to no PH group (while the other two PH groups showed significantly worse survival compared to no PH group). They concluded that novel PH definition has prognostic significance. This paper is well-written and contains important clinical implications. However, there are a number of flaws and pitfalls that should be addressed to improve the quality of the manuscript.

Major points.

1.      The novel definition of PH lowers the mPAP threshold of PH from 25 mmHg down to 20 mmHg. If this new definition of PH is to work for the risk stratification of the patients with aortic stenosis (AS) undergoing AVR, patients in the gray zone (i.e. mPAP: 21-24 mmHg), who would have been diagnosed as having no PH by previous definition, should show higher 1 year mortality compared to the ‘no PH group’. However, the 3rd figure in Figure 3 (survival according to mPAP categories) shows no difference in survival between the mPAP 21-24 mmHg group (n=51) and the mPAP £ 20 mmHg group (n=134). Given the number of patients in each group, I presume the mPAP 21-24 mmHg group (as well as mPAP ³ 25 mmHg group) may include patients whose elevated mPAP are deemed ‘hyperdynamic’ (i.e. no PH with elevated mPAP). In this regard, this figure is misleading and would be better omitted. What really intriques me is whether this borderline patients also have a higher risk for death after TAVR compared to no PHT group. Thus, my questions are: 1) How many patients in the mPAP 21-24 mmHg group were finally categorized as having PH, excluding hyperdynamic patients? 2) Is the one year survival of this real borderline cases different from that of no PH group? The authors should add a new figure to show the outcomes of this newly diagnosed patients with marginal PH in comparison to those of the ‘no PHT’ group so that they could prove their hypothesis that new definition of PH is beneficial for teasing out more the borderline cases of PH who used to be diagnosed as having no PH by previous definition.

2.      The other conclusion of this study - Survival of IpcPH group was comparable to no PH group – sounds sensible. However, the authors should explain why, providing appropriate data. The pathogenesis of the development of PH in patients with AS may be 1) AS with compensating LV and no or mild elevation of the left atrial pressure (LAP), and, as a result, no PH, 2) AS with early LV decompensation, significant elevation of LAP, and, as a result, significant PH, but neither pulmonary vascular change nor elevation of pulmonary arterial resistance (PAR) (IpcPH), 3) AS with late LV decompensation, significant elevation of LAP and concomitant elevation of PAR, and significant PH (CpcPH). Pre-cap PH may be a result of either the association of lung diseases or overly elevated mPAP in response to mild elevation in LAP (out-of-proportion PH). Table 2 summarize the pre-TAVR echocardiographic data, showing the differences among these four groups, which appears sensible. If patients in no PH group or IpcPH group show better survival and LV recovery after TAVR compared to the rest of the patients, so do their follow-up echocardiographic data (TR velocity as a surrogate for the severity of residual PH, parameters for left ventricular diastolic function, LV geometry, LV mass index…). I would like the authors to provide post-TAVR echocardiographic findings, comparing the data among the groups, and hopefully showing better echocardiographic findings in the no PH or IpcPH group, asserting that better survival in this two groups is attributed to the better recovery of LV after TAVR.

Minor points.

1.      It is inappropriate to put in Cox regression results to the Kaplan-Meier curves (Figure 3,4) because the latter is unadjusted comparison. Log rank test is usually used to ascertain intergroup difference.  

2.      In Figure 1 second row, ‘PH’ should be changed to ‘no PH’, and ‘all’ should be omitted.

3.      In calculating the percentage of the patients in the subgroups, different denominators should be described. For instance, line 193 should be ‘… in the IpcPH (41/78, 53%) but less frequent in the CpcPH (27/82, 33%)…’.

4.      Post capillary should be hyphenated (post-capillary) if pre-capillary is to be hyphenated.

5.      The last sentence in the abstract ‘Patients with a pre-capillary PH component present an even worse prognosis’ would be better to be changed as ‘Prognosis of isolated post-capillary PH was comparable to that of no PH’, because the latter finding seems more important the former.  

Author Response

Please see attachement.

Reviewer 2 Report

In this paper, authors described a prognosis in patients after Transcatheter Valve Replacement (TAVR) with a novel pulmonary hypertension definition. According to the new definition of pulmonary hypertension recommended in 6th World Symposium of Pulmonary hypertension, combination of pulmonary hypertension before TAVR still remains an independent predictor of prognosis after intervention. They also explained that complication of pre-capillary disease (Pre-cap PH and CpcPH) showed poor prognosis compared to that of No PH patients. The manuscript is well written; however, I have some comments for this article.

Major Concerns:

1.      L81-84: In this study, 40 patients with a mPAP>20mmHg, a normal range of PAWP and a PVR were considered with no PH due to hyper-dynamic state during right heart catheterization. However, these patients should be diagnosed as PH because their pulmonary artery pressure were elevated over 20mmHg. I recommend you to exclude these patients from no PH.

2.      L322-326: The discussion section should be more focused on the cause of no significant difference between patients with mPAP 21-24 mmHg and mPAP20 mmHg if authors expect patients with mPAP 21-24mmHg have a poor prognosis. Although new criteria of pulmonary hypertension were related to poor outcome, the importance of using these criteria is unclear, especially in patients with mPAP 21-24mmHg. It means that authors should explore the clinical significance of new criteria compared to old criteria.

3.      In multivariate analysis, authors included only baseline EuroScore II as an independent covariate. However, EuroScore II is a risk estimation of in-hospital death after cardiac surgery. It has been reported that mortality in patients with low-flow low-gradient AS was higher than that of high-gradient AS patients after TAVR (JACC Cardiovasc Interv.2019 Apr 22;12(8):752-763.). Authors have to include more covariates associated with prognosis after TAVR.

 Minor concerns:

1.      L294: hmodynamics→hemodynamics.

Round 2

Reviewer 1 Report

Thank you for your efforts to revise the manuscript. I would like to acceopt the paper as it is.

Reviewer 2 Report

Authors described all my concerns, and now I feel this article is suitable for Journal of Clinical Medicine.